# Genomics of Preaxostyla Flagellates Illuminates the Path Towards the Loss of Mitochondria

Lukáš V. F. Novák[1,2]*, Sebastian C. Treitli[1,3], Jan Pyrih[1], Paweł Hałakuc[4], Shweta V. Pipaliya[5,6], Vojtěch Vacek[1], Ondřej Brzoň[1], Petr Soukal[1], Laura Eme[7], Joel B. Dacks[5,8], Anna Karnkowska[4], Marek Eliáš[9], Vladimír Hampl[1]*

1 Charles University, Faculty of Science, Department of Parasitology, BIOCEV, Vestec, Czech Republic, 2 Université de Bretagne Occidentale, CNRS, Unité Biologie et Ecologie des Ecosystèmes Marins Profonds BEEP, IUEM, Plouzané, France, 3 RG Insect Gut Microbiology and Symbiosis, Max Planck Institute for Terrestrial Microbiology, Marburg, Germany, 4 Institute of Evolutionary Biology, Biological and Chemical Research Centre, Faculty of Biology, University of Warsaw, Poland, 5 Division of Infectious Diseases, Department of Medicine, University of Alberta, Edmonton, Canada, 6 School of Life Sciences, École Polytechnique Fédérale de Lausanne, Lausanne, Switzerland; Swiss Institute of Bioinformatics, Lausanne, Switzerland, 7 Ecology, Systematics, and Evolution Unit, Université Paris-Saclay, CNRS, Orsay, France, 8 Institute of Parasitology, Biology Centre, Czech Academy of Sciences, České Budějovice, Czechia, 9 University of Ostrava, Faculty of Science, Department of Biology and Ecology, Ostrava, Czech Republic

* lukas.novak@univ-brest.fr (LVFN); vlada@natur.cuni.cz (VH)

**Data Availability Statement:** The annotated genomes were deposited in NCBI under accession number JAPMOS000000000 for P. pyriformis and JARBJD000000000 for B. nauphoetae.

## Abstract

The notion that mitochondria cannot be lost was shattered with the report of an oxymonad *Monocercomonoides exilis*, the first eukaryote arguably without any mitochondrion. Yet, questions remain about whether this extends beyond the single species and how this transition took place. The Oxymonadida is a group of gut endobionts taxonomically housed in the Preaxostyla which also contains free-living flagellates of the genera *Trimastix* and *Paratrimastix*. The latter two taxa harbour conspicuous mitochondrion-related organelles (MROs). Here we report high-quality genome and transcriptome assemblies of two Preaxostyla representatives, the free-living *Paratrimastix pyriformis* and the oxymonad *Blattamonas nauphoetae*. We performed thorough comparisons among all available genomic and transcriptomic data of Preaxostyla to further decipher the evolutionary changes towards amitochondriality, endobiosis, and unstacked Golgi. Our results provide insights into the metabolic and endomembrane evolution, but most strikingly the data confirm the complete loss of mitochondria for all three oxymonad species investigated (*M. exilis*, *B. nauphoetae*, and *Streblomastix strix*), suggesting the amitochondriate status is common to a large part if not the whole group of Oxymonadida. This observation moves this unique loss to 100 MYA when oxymonad lineage diversified.

## Author summary

Mitochondria are nearly ubiquitous components of eukaryotic cells that constitute bodies of animals, fungi, plants, algae, and a broad diversity of single-celled eukaryotes, a.k.a.

**Funding:** This project has received funding from the European Research Council (ERC) under the European Union's Horizon 2020 research and innovation programme (grant agreement No. 771592 to VH) and the Centre for research of pathogenicity and virulence of parasites (registration no. CZ.02.1.01/0.0/0.0/16_019/0000759). Research in Karnkowska lab was supported by EMBO Installation Grant 4150 and Ministry of Education and Science, Poland and the Interdisciplinary Centre for Mathematical and Computational Modelling (ICM) University of Warsaw under computational allocation no. G 72-16. Research in the Dacks Lab is supported by grants from the Natural Sciences and Research Council of Canada (RES0021028, RES0043758, and RES0046091) and SVP received salary support through Alberta Innovates Graduate Studentship (Doctoral) and Canadian Institutes of Health Research Canada Graduate Scholarships. LE was supported by an ERC Starting grant (803151). ME was supported by the Czech Science Foundation project 22-29633S. LVFN, SCT, JP, VV, OB, PS, and VH received a salary from ERC (771592). ME received a salary from the Czech Science Foundation (22-29633S). The funders had no role in study design, data collection and analysis, decision to publish, or preparation of the manuscript.

**Competing interests:** The authors declare that they have no conflict of interest.

protists. Many groups of protists have substantially reduced the complexity of their mitochondria because they live in oxygen-poor environments, so they are unable to utilize the most salient feature of mitochondria–their ATP-producing oxidative phosphorylation metabolism. However, for a long time, scientists thought that it is impossible to completely lose a mitochondrion because this organelle provides other essential services to the cell, e.g. synthesis of protein cofactors called iron-sulfur clusters. Detailed investigation of the chinchilla symbiont *M. exilis* documented the first case of an organism without mitochondrion, and it also provided a scenario explaining how this unique evolutionary experiment might have happened. In this work, we expand on this discovery by exploring genomes of multiple relatives of *M. exilis*. We show that the loss of the mitochondrion is not limited to a single species but possibly extends to its entire group, the oxymonads. We also compare the predicted metabolic capabilities of oxymonads to their closest known mitochondrion-containing relatives and map out various changes that occurred during the transition to amitochondriality.

## Introduction

Multiple eukaryotic lineages have adapted to low-oxygen and/or endobiotic lifestyles by modifying their mitochondria into a wide range of mitochondrion-related organelle (MRO) types via the process of reductive and adaptive evolution [1]. The most radically modified MROs are traditionally categorized as hydrogenosomes, producing ATP by extended glycolysis, and as mitosomes with no role in energy metabolism [2]. However, exploration of a broader diversity of MRO-containing lineages makes it clear that MROs of various organisms form a functional continuum [3–6]. The extreme outcome of mitochondrial reductive evolution is the complete loss of the organelle, but so far only one organism has been conclusively shown to have reached this point, the chinchilla gut symbiont *Monocercomonoides exilis* (Oxymonadida, Preaxostyla, Metamonada). A genomic project has thoroughly corroborated the amitochondriate status of *M. exilis*, in which it failed to identify any mitochondrion-associated genes while showing multiple other eukaryotic cellular systems to be well represented [7–9].

Oxymonadida contains approximately 140 described species of morphologically divergent and diverse flagellates exclusively inhabiting the digestive tracts of metazoans, of which none has been conclusively shown to possess a mitochondrion by cytological investigations [10]. Therefore, the entire Oxymonadida group may be an amitochondriate lineage, with the mitochondrion being lost in its common ancestor. Oxymonads belong to Preaxostyla, one of the five major clades of the phylum Metamonada consisting of exclusively anaerobes or microaerophiles; the other clades are represented by Fornicata (e.g., *Giardia intestinalis*), Parabasalia (e.g., *Trichomonas vaginalis*), barthelonids and their relatives (the BRC group; [11,12]), and Anaeramoebidae (e.g., *Anaeramoeba flamelloides*; [13]). The two additional known branches of Preaxostyla, classified as the genera *Trimastix* and *Paratrimastix*, split out at two successive points off the trunk leading to Oxymonadida (Fig 1A). They are both comprised of free-living, bacterivorous flagellates exhibiting a typical excavate morphology and ultrastructure and thriving in low-oxygen environments [14].

*Paratrimastix pyriformis* has been shown to possess an MRO morphologically resembling a hydrogenosome. However, it is likely not involved in the ATP-generating extended glycolysis but plays a role in the one-carbon metabolism of the cell [15–17]. Putative MROs have also been observed in electron microscopy studies of a *Trimastix* representative, *T. marina* [14], and their nature was illuminated by transcriptomic data [5]. As a group with at least two

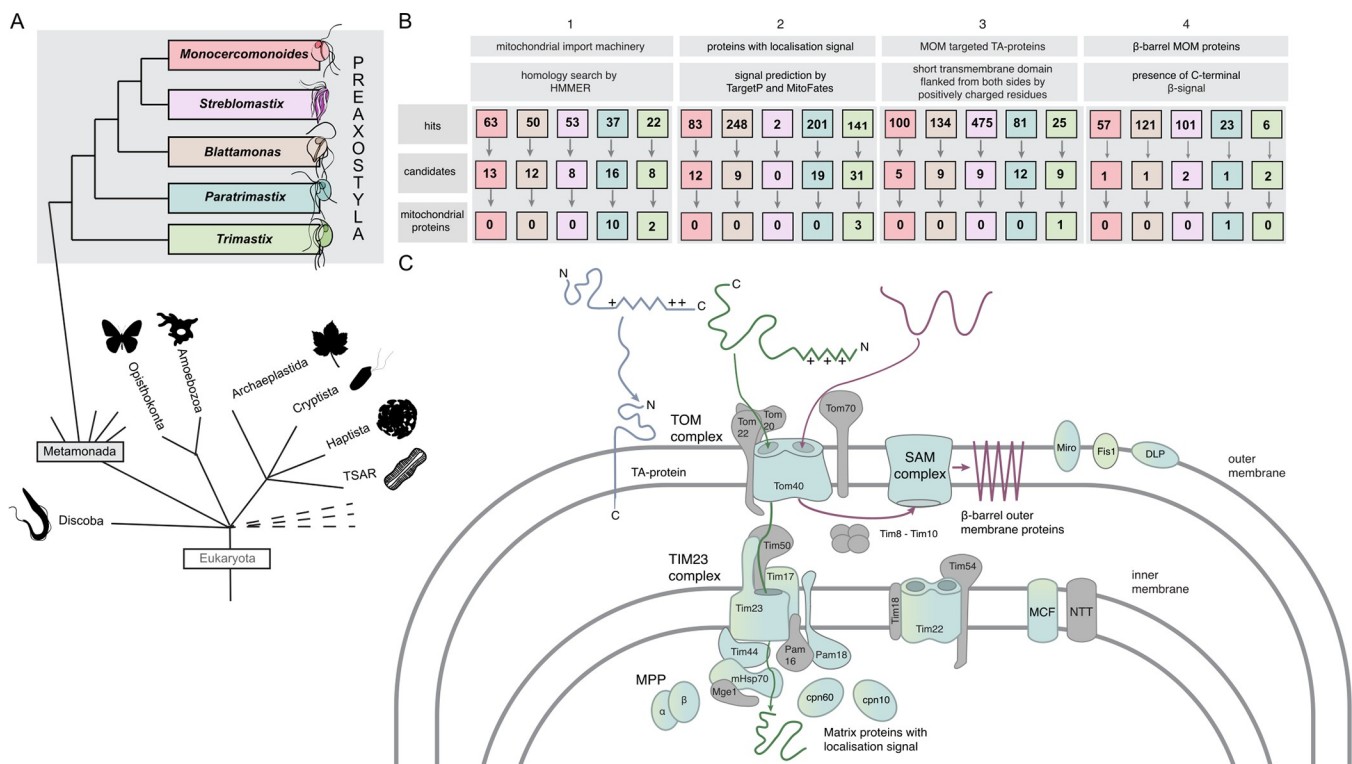

**Fig 1. Summary of the searches for proteins physically associated with MROs.** (A) Schematic representation of the position of Preaxostyla in the eukaryotic tree of life and the topology within the group. Representative non-preaxostylan organism silhouettes are from PhyloPic (phylopic.org). (B) Results of four searches for putative mitochondrial proteins are summarized and species are color-coded as in A. Detailed information on the three steps of the searches and candidates is given in S2 File. (C) Mitochondrion hallmark proteins detected in the data sets of *P. pyriformis* and *T. marina* are shown in blue/green. No such candidate was recovered for any oxymonad representative. Conserved mitochondrial protein not detected in any studied preaxostylan organism are shown in grey.

MRO-bearing lineages and an amitochondriate species/clade nested within, the Preaxostyla clade provides a promising model system to study the causes, conditions, and consequences of the loss of this ubiquitous cellular compartment.

To start answering questions about the timing and circumstances of this single evolutionary experiment in which the mitochondrion was lost from the cell, a denser sampling of omics-level data from Preaxostyla is needed. Currently, the available data encompasses a genome draft of *M. exilis*, transcriptomes of *P. pyriformis* and *T. marina* with variable completeness, and a fragmentary single-cell genome assembly of an oxymonad *Streblomastix strix*. As a key step towards this end, we present high-quality genomic assemblies for another oxymonad, *Blattamonas nauphoetae*, and for the free-living *P. pyriformis*. The distribution of mitochondrion hallmark proteins and comparisons of gene repertoires and metabolic functions among five Preaxostyla species of various ecology and MRO status were used to illuminate the adaptations connected to amitochondriality, loss of stacked Golgi, and the origin of endobiotic lifestyle within the group.

## Results and discussion

### The genome assemblies of *Paratrimastix pyriformis* and *Blattamonas nauphoetae* are highly contiguous and nearly complete

We employed a combination of Oxford Nanopore and Illumina technologies to obtain genome assemblies of two species of Preaxostyla. As both species are grown in polyxenic

**Table 1. General features of the Preaxostyla genomes discussed in this study.**

| Sample | Scaffolds | Total length (bp) | N50 (kbp) | Completeness (%; BUSCO odb9) | Completeness (%; Merqury) | G+C content (%) | Protein-Coding Loci | Source |
|---|---|---|---|---|---|---|---|---|
| *P. pyriformis* | 633 | 56,627,582 | 276.6 | 76.3 | 94.01 | 60.94 | 10,815 | NCBI JAPMOS000000000 |
| *B. nauphoetae* | 879 | 88,537,989 | 199.5 | 76.6 | 98.24 | 44.96 | 25,221 | NCBI JARBJD000000000 |
| *M. exilis* | 2,092 | 74,712,536 | 71.44 | 75.3 | n/a | 36.8 | 16,768 | GiardiaDB release 46 GiardiaDB-46_MexilisPA203 |
| *S. strix* | 50,889 | 152,152,197 | 5.18 | 69.6 | n/a | 26.6 | 56,706 | NCBI SNRW00000000 |

cultures where the eukaryotes represent a minority of cells, we employed multiple rounds of decontamination. Prior to the genomic DNA isolation, enrichment of the sample for the eukaryotic cells was achieved using filtration, and after sequencing the data were carefully decontaminated bioinformatically, resulting in two highly contiguous eukaryotic genome assemblies (see Materials and methods). The basic characteristics of these genomic assemblies and their comparison to the previously published assemblies of Preaxostyla taxa are given in Table 1.

The *B. nauphoetae* genome was assembled into 879 contigs spanning 88,537,989 bp, with an N50 = 199,589 bp and a GC content of 44.96%. Automatic and manual gene prediction resulted in 25,221 predicted protein-coding genes. The *P. pyriformis* genome was assembled into 633 scaffolds spanning 56,627,582 bp, with an N50 = 276,605 bp and a GC content of 60.94%. Manual and automatic gene prediction resulted in 10,815 predicted protein-coding genes.

Using BUSCO v3 [18] with the eukaryota_odb9 dataset, the genome completeness was estimated to be 76.3% and 76.6% for *P. pyriformis* and *B. nauphoetae*, respectively (Table 1). BUSCO values, despite their wide usage, are not expected to reach 100% in lineages distant from model eukaryotes simply due to the true absence or high sequence divergence of some of the assessed marker genes. A detailed list of BUSCO markers in Preaxostyla and a comparison of their presence/absence in high-quality genomes of *Trypanosoma brucei*, *Giardia intestinalis*, and *Trichomonas vaginalis* is given in Sheet A in S1 File. It demonstrates that the absence of many BUSCO markers is generally shared by the Preaxostyla species indicating that this is true absence of the given gene in the genome rather than reflection of data incompleteness. To avoid a bias caused by evolutionary divergence, we also attempted to estimate the genome completeness using reference-independent approaches with Merqury [19]. Using this method, we obtained completeness values of 94.01% and 98.24% for *P. pyriformis* and *B. nauphoetae*, respectively (Table 1). Finally, we also searched for 79 conserved ribosomal proteins [20]. We identified 74 of them in the predicted gene set of *P. pyriformis*, with the remaining five identified by manual searches in the genome (Sheet B in S1 File). A comparable result was achieved for *B. nauphoetae*, where we identified 76 of the ribosomal proteins in the predicted gene set, and the remaining three were identified by a manual search in the genome (Sheet C in S1 File). Altogether our analyses suggest that the *P. pyriformis* and *B. nauphoetae* genome drafts are highly complete.

## The absence of mitochondrial hallmark genes in oxymonads extends beyond *M. exilis*

The hypothesis of the mitochondrion organelle absence in *M. exilis* was postulated based on the absence of mitochondrion-related proteins in its genome and transcriptome and led to

overturning of the paradigm that mitochondria are ubiquitous among eukaryotes [7]. Obviously, the amitochondriate status of *M. exilis* is immediately falsifiable by finding any evidence of a putative organelle in this species. Careful inspection of the predicted proteomes of other oxymonads for the presence of mitochondrial hallmarks is an obvious next step that may further support, or weaken, this hypothesis. To follow this line of investigation, we have carefully searched the genome drafts of *P. pyriformis* and *B. nauphoetae* published in this study, the previously published genome drafts of *Monocercomonoides exilis* [7,21] and *Streblomastix strix*, and the previously published transcriptome of *Trimastix marina* [5], i.e., a composite dataset containing three oxymonad species and two other free-living Preaxostyla, the latter clearly possessing an MRO [5,17].

We used several methods to identify candidate mitochondrial proteins (results for all searches are shown in S2 File and summarized in Fig 1B and 1C). In most cases, proteins identified in the first step were searched against the custom mitochondrial protein sequence database based on the MitoMiner database [22,23] (herein referred to as MitoMiner) and those with hits were considered as candidate mitochondrial proteins. In the last step the functional annotation of the candidates based on NCBI best hits or additional hint, co-occurrence across Preaxostyla, were considered to select the putative mitochondrial proteins from the candidates. For the purpose of the analyses the predicted proteins of Preaxostyla were clustered with predicted proteins of other Metamonada into orthologous group (S3 File).

As the most obvious approach, we searched all five investigated species for homologues of nuclear genome-encoded proteins typically associated with mitochondria or MROs in other eukaryotes including *Paratrimastix* and *Trimastix*. In the first step, we used profile Hidden Markov Models (HMMs) to search for components of the mitochondrion protein import and maturation machinery, considered one of the most conserved mitochondrial features [24]. Homology searches resulted in 22 (*T. marina*) to 63 (*M. exilis*) hits, which were further evaluated by searches against the MitoMiner resulting in 57 candidates in total. Functional annotations indicated that all candidates recovered in oxymonad data sets are very likely false positives: mainly vaguely annotated kinases, peptidases, and chaperones (Fig 1B and S2 File). For *P. pyriformis* and *T. marina*, the situation was different. Out of the 11 previously identified components of the *P. pyriformis* mitochondrion protein import and maturation machinery [16,17], 10 were found in the predicted proteomes of these two species with this approach: the β-barrel outer membrane translocases Tom40 and Sam50 and the associated receptor Tom22, the inner membrane translocases Tim17, Tim22 and their associated protein Pam18 and Tim44, the α and β subunit of the mitochondrial processing peptidase (MPP), and the mitochondrial chaperone protein HSP70. Tim17 and mtHsp70 were also identified in the *T. marina* dataset, corroborating previous findings [5]. The only exception was the previously detected Tim23 protein (PAPYR_1418; [17]), which was missed in our new search because of its high sequence divergence. The successful identification of the protein translocon components in these two species validated our approach.

The sensitive HMM searches were complemented by an extensive search for putative homologues of known mitochondrial proteins using a pipeline based on the MitoMiner database (S2 File). As already shown for *M. exilis*, the specificity of the pipeline in organisms with divergent mitochondria is low [7]. The search recovered a similar number of hits in all investigated species, from 636 in *S. strix* up to 1,024 in *P. pyriformis*. An objective assessment of the localization of all these proteins is not realistic. Many of them are clearly not mitochondrial (e.g., histones or ribosomal proteins), and others belong to very general gene ontology categories (e.g., protein-binding) making the assessment impossible. After individually inspecting the remaining items, we conclude there is none among the hits from oxymonads that would raise a strong case for the presence of mitochondria. On the other hand, reliable mitochondrial

proteins were found among hits from *P. pyriformis* and *T. marina*. The first such examples are dynamin-like proteins (PAPYR_3413 and Gene.668::gnl|Trimastix_PCT|268). Their phylogenetic position (Fig A in S4 File) corroborates the hypothesis of Karnkowska et al. [8] that these may mediate the division of MROs in these species. Other examples are proteins (PAPYR_2826 and Gene.3674::gnl|Trimastix_PCT|1191) orthologous to the mitochondrial outer membrane-anchored protein MIRO (mitochondrial Rho; [25]) broadly conserved in eukaryotes but rare in MRO-containing taxa [26,27]. Indeed, PAPYR_2826 was recently confirmed by proteomics to be MRO-associated in *P. pyriformis* [17]. Finally, the search recovered homologues of mitochondrial matrix chaperones Cpn60 and Cpn10 and homologues of mitochondrial carrier family proteins used for export or import ATP and other metabolites [28]. Bacterial-type (NTT-like) nucleotide transporters [29], sporadically used for ATP transport, were not recovered neither by this search nor by targeted BLAST searches.

## Homology-independent searches do not identify convincing candidates for mitochondrial proteins in the sequenced oxymonad genomes

As an alternative to homology searches, we also searched our Preaxostyla sequence datasets for several types of signature sequences typical for mitochondrion-targeted proteins. The matrix proteins of mitochondria and MROs are expected to contain characteristic N-terminal targeting signals (NTS) needed for the targeted import into MROs [30]. However, it has been previously recognized that the presence of a predicted NTS by itself does not prove the targeting, as such amino acid sequences can also appear by chance [31]. Indeed, we have previously shown that 14 *M. exilis* proteins are imported into hydrogenosomes when heterologously expressed in *T. vaginalis* [32]. Here we used TargetP v1.1 and MitoFates v1.1 for mitochondrial targeting signal prediction and identified from as little as two hits in *S. strix* up to 248 hits in *B. nauphoteae* (S2 File). Only 21 of these oxymonad sequences found hits in MitoMiner rendering them candidates which were considered further and annotated by BLAST searches against the NCBI nr database (Fig 1B). None of these annotations strongly suggest a mitochondrial function. As an additional hint, we considered the conservation of NTS prediction across Preaxostyla. We reasoned that if the NTS in the protein is real it should be detected also in orthologues from other Preaxostyla species. Only four proteins from oxymonads (ribosomal protein L21, L23a, L34e, and tRNA-dihydrouridine synthase) fulfilled this criterion, none of them representing a reasonable mitochondrial protein in organisms that certainly lack a mitochondrial genome. Based on these results, we assume that all NTS predictions on oxymonad proteins are false positives. Altogether, 50 of *P. pyriformis* and *T. marina* proteins with predicted NTS found hits in MitoMiner rendering them candidates. This probably reflects the presence of the organelle. Indeed, we identified among them, for example, aminoadipate-semialdehyde dehydrogenase, GCS-H, and mtHSP70 proteins all previously suggested to localize to the MRO [5].

The outer mitochondrial membrane accommodates two special classes of proteins, TA proteins and mitochondrial β-barrel membrane proteins (MBOMPs), the former using specific C-terminal signals [33–35]. We identified up to 475 TA hits in the predicted proteome of *S. strix* and around 100 for the rest of the species, with only 25 for *T. marina* (S2 File). Depending on the species only five to 12 were considered as mitochondrial candidates (i.e. having hits in the MitoMiner database) (Fig 1B) and the majority of these were Golgi apparatus/ER-related upon closer scrutiny. Like in the case of NTS-containing proteins, we assessed if the set of predicted TA proteins contains groups of orthologues containing more than a single Preaxostyla species. Indeed, 22 such orthologous groups were found. These were mostly without MitoMiner hits and in most cases annotated as SNARE or other proteins involved in vesicle trafficking outside mitochondria. This suggests that the TA prediction did produce true positives, but they most

likely represent non-mitochondrial TA proteins. The only robust mitochondrial candidate from the whole list was Fis1 (mitochondrial fission protein 1) in *T. marina*, which is a tail-anchored (TA) protein mediating mitochondrial fission [36].

Our search for proteins with MBOMP characteristics revealed six and 23 hits in *T. marina* and *P. pyriformis*, respectively, but only three of them had a hit in the MitoMiner database, including the *P. pyriformis* Tom40, the detection of which validated our approach (Fig 1B). Among the hits from the oxymonad genomes, only four hit a protein in the MitoMiner database rendering them mitochondrial candidates, none of them being a known MBOMP (S2 File). Only 10 orthologous groups predicted as MBOMPs contained representatives of more than one Preaxostyla species. The annotations of none of them suggested a mitochondrial function: clathrin heavy chain, kelch-type beta-propeller, EF-hand domain-containing protein, eukaryotic translation initiation factor 3 subunit B, and BTB/POZ domain-containing protein. The only positive case from this set thus remained Tom40 in *P. pyriformis*.

### Probing the oxymonad genomes with the *T. brucei* reference mitochondrial proteome provides further support for the absence of mitochondria in oxymonads

Given the limitations of the previous searches, namely, high false-positive rates and uncertainty about mitochondrial localization of proteins included in the MitoMiner database, we have performed yet another analysis in which we used a carefully curated set of queries. In this analysis, we searched for Preaxostyla orthologues of proteins from the experimentally well-established mitochondrial proteome of *Trypanosoma brucei* [37–40]. The selection of this organism as a reference also considered the fact that it is a member of the eukaryotic supergroup Discoba, which frequently forms a sister group to Metamonada in phylogenomic analyses [41], making *T. brucei* potentially the closest relative of metamonads out of all eukaryotes with extensively studied mitochondria. Reciprocal BLAST searches of the *T. brucei* mitochondrial proteins against the predicted proteins of Preaxostyla revealed ~200 putative groups of orthologues and those were investigated manually. Careful inspection of the raw localization data in *T. brucei* and protein phylogenies rejected most cases by disputing either the mitochondrial localization in *T. brucei* or the orthology; the latter was considered uncertain if *T. brucei* and Preaxostyla proteins were separated by prokaryotic or non-mitochondrial eukaryotic homologues in the phylogenetic trees (S5 File). The 30 cases passing this initial scrutiny (S2 File) were divided into two groups.

The high-confidence group comprises 17 proteins that likely share mitochondrial origin as they formed a monophyletic cluster with other mitochondrial homologues. These were present only in *P. pyriformis* and/or *T. marina* and never in oxymonads. Out of these 17 proteins, 12 were already predicted to be mitochondrial previously [17]. The remaining proteins, such as L-threonine 3-dehydrogenase (TDH) or glycine acetyltransferase, are thus promising novel candidates for mitochondrial proteins in these two protists.

The 13 putative mitochondrial proteins in the low confidence group clustered with known mitochondrial proteins or proteins predicted to reside within the mitochondrion using the TargetP v1.1 algorithm, although the support for the mitochondrial targeting of proteins within these clusters was not consistent. All such clusters also included cytosolic or peroxisomal isoforms or the enzymes known to be dually localized, which questions the mitochondrial origin of these clusters. An example of the former is isocitrate dehydrogenase, for which the yeast cytosolic, peroxisomal, and mitochondrial proteins form a clade in the tree (S5 File). Examples of the latter are aconitase and alanine aminotransferase, for which a dual cytosolic/mitochondrial location was reported for various eukaryotes including trypanosomatids

[42,43]. This low confidence group contained nine proteins present also in oxymonads (S2 File), and these are discussed in more detail below.

Aspartate and alanine aminotransferases and the two tRNA synthetases in this category are all dually localized in the mitochondrion and the cytosol of *T. brucei*. In the case of the tRNA synthetases, the dual localization arose specifically in trypanosomatids [39] and, consistently with this hypothesis, Preaxostyla and *T. brucei* proteins cluster together with cytosolic tRNA synthetases of other eukaryotes. In the case of aspartate aminotransferase, the dual localization seems to be of a deeper evolutionary origin [44]. The two mevalonate pathway enzymes (hydroxymethylglutaryl-CoA synthase and 3-hydroxy-3-methylglutaryl-CoA reductase) are localized in the mitochondrion of kinetoplastids [45], however, this is a specialty of this line-age, as eukaryotes typically run this pathway in the cytosol/ER. In all these cases, we conserva-tively assume that these enzymes were localized in the cytosol in the common ancestor of Preaxostyla and *T. brucei* and were partially or fully moved to the mitochondrion in the lineage leading to *T. brucei*. The last candidate was the malic enzyme. There are two homologues of the enzyme in kinetoplastids, one cytoplasmic and one mitochondrial, which cluster together (Fig B in S4 File). Again, it cannot be inferred what the situation in the common ancestor with Preaxostyla was, which does not give a strong reason to assume MRO localization in Preaxostyla.

In summary, our systematic searches for MRO proteins allowed us to update the predicted set of MRO-localized proteins of *P. pyriformis* and *T. marina* (S6 File). Most critically, no reli-able candidate for an MRO protein was detected in any of the oxymonad data sets, supporting the hypothesis of the absence of the mitochondrion in *M. exilis* and, importantly, extending the amitochondriate status to two other oxymonad species. After this new revelation, we won-dered how the loss of mitochondria in the three oxymonads species had seeped into functions related to this organelle, namely: extended glycolysis, amino acid metabolism, the complement of FeS cluster-containing proteins, heme synthesis, and peroxisomes.

## Inventory of enzymes for extended glycolysis shows a richer diversity of FeFe hydrogenases in *Trimastix* and *Paratrimastix* than in oxymonads

Many eukaryotic anaerobes, including Preaxostyla, generate ATP using the extended glycolysis pathway, which produces acetate, $CO_2$, and $H_2$ from pyruvate while performing substrate-level phosphorylation of ADP to ATP [46]. The pathway uses pyruvate as a substrate, which is either directly sourced from glycolysis or produced by decarboxylation of malate through the activity of the malic enzyme (ME), which was identified in all five Preaxostyla species. All studied Pre-axostyla species apparently rely on oxidative decarboxylation of pyruvate to acetyl coenzyme A (acetyl-CoA) and $CO_2$ in a reaction catalyzed by pyruvate:ferredoxin oxidoreductase (PFO; [47]), owing to the identification of three to six PFO isoforms in each species analyzed (Figs 2 and C in S4 File and S6 File). None of the alternative enzymes mediating the conversion of pyruvate to acetyl-CoA, pyruvate:NADP$^+$ oxidoreductase (PNO) and pyruvate formate lyase (PFL), could be detected in any of the studied species.

Both the decarboxylation of malate by ME and of pyruvate by PFO are oxidative processes that release electrons, producing NADH and reduced ferredoxin, respectively, and these elec-tron carriers need to be reoxidized. The final fate of the electrons carried by ferredoxin often lies in the reduction of protons to molecular hydrogen through the activity of [FeFe] hydroge-nases (HydA; [48]). In addition to the "simple" hydrogenases, which are present in all species of Preaxostyla, [FeFe] hydrogenases with N-terminal homology to the NuoG subunit of NADH-quinone oxidoreductase and a C-terminal homology to NADPH-dependent sulfite reductase (CysJ), were identified in the MRO-containing *T. marina* and *P. pyriformis* (Figs 3

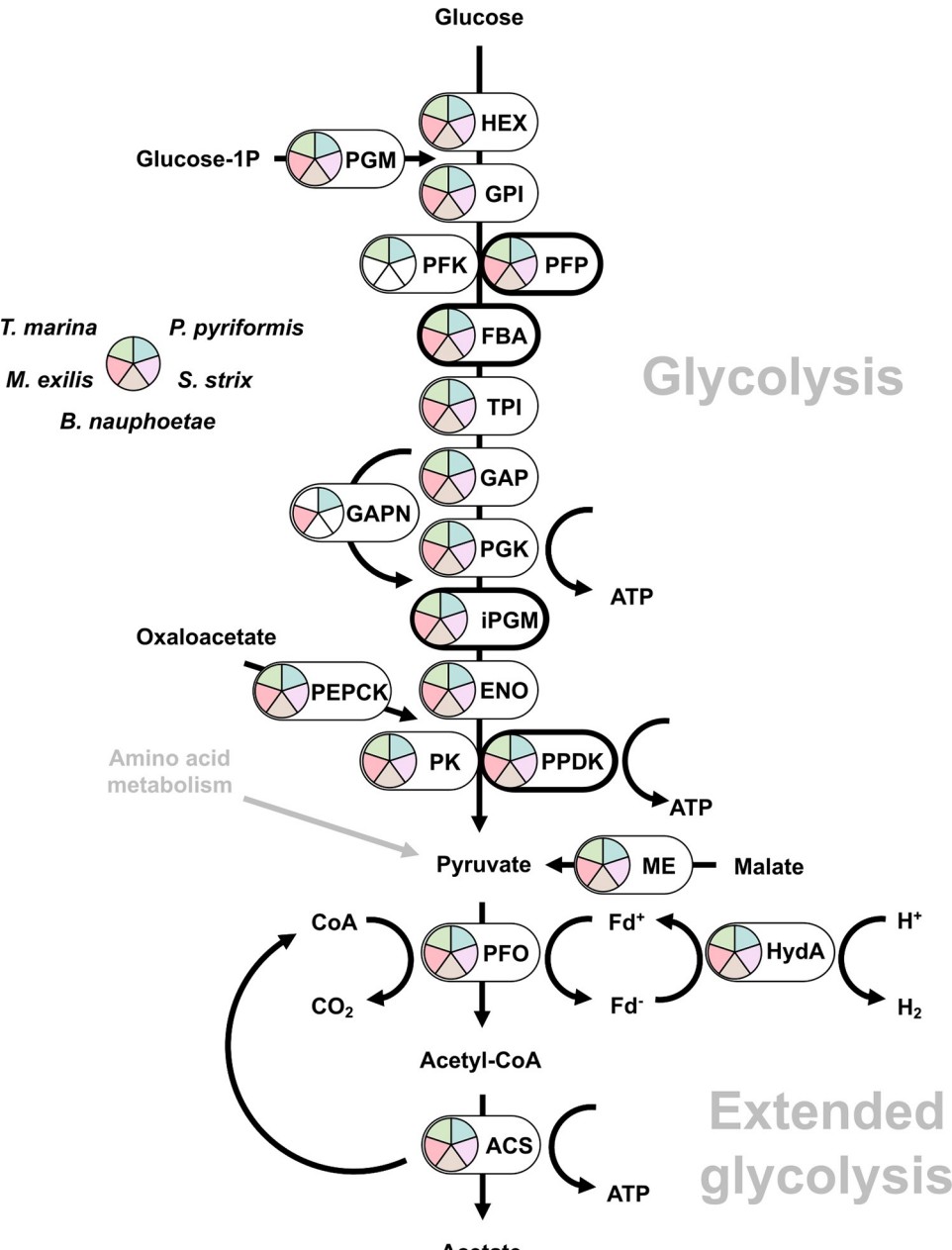

**Fig 2. Hypothetical reconstruction of the extended glycolysis in Preaxostyla.** Bold outline indicates alternative glycolytic enzymes. Abbreviations and Enzyme Commission numbers are given in S6 File. Presence of the enzymes in Preaxostyla data sets is indicated by a color code.

and D in S4 File). Similar "fused" hydrogenases have been previously reported in other eukaryotic anaerobes, including *T. vaginalis* [49], the breviate *Pygsuia biforma* [50], the jakobid *Stygiella incarcerata* [51], and the amoebozoan *Pelomyxa schiedti* [52]. Although they do not belong to the group of A3 trimeric hydrogenases [53] known to be capable of NADH oxidation via electron confurcation [54], they were hypothesized to catalyze NAD(P)H-dependent formation of $H_2$ [49]. Although the presence of these large hydrogenases correlates with the presence of MROs, they were not detected in the MRO proteome of *P. pyriformis* [17] and their localization is unknown.

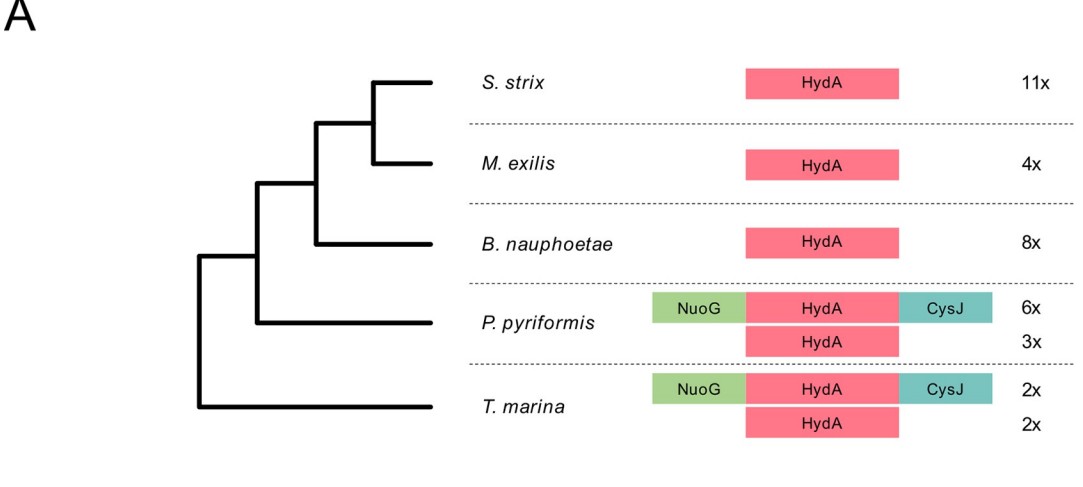

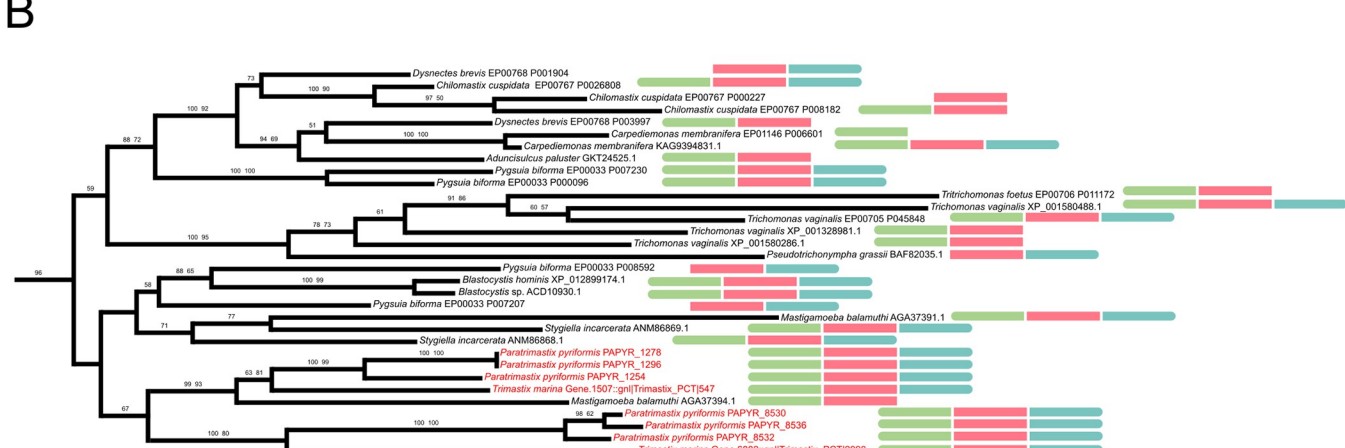

**Fig 3. Phylogenetic relationship among hydrogenases of Preaxostyla.** (A) Schematic representation of domain architectures of hydrogenases in Preaxostyla species. HydA: [FeFe] hydrogenase; CysJ: NADPH-dependent sulfite reductase; NuoG: NADH-quinone oxidoreductase. (B) Detailed view of the part of a hydrogenase phylogenetic tree that corresponds to a clade including long hydrogenases. The domain architecture of the proteins is indicated by color bars. The full tree is given in Fig D in S4 File.

The last part of the extended glycolysis, which yields ATP, acetate, and CoA, can be catalyzed either by a two-enzyme system consisting of acetate:succinate CoA-transferase (ASCT) and succinyl CoA synthetase (SCS) like in *T. vaginalis*, or by a single enzyme acetyl-CoA synthetase (ACS) like in *G. intestinalis*. All five Preaxostyla species contain ACS, while neither ASCT nor SCS was identified. ACS has a complicated evolutionary history in Metamonada characterized by multiple LGT events and gene losses [5,11,55]. The majority of ACS homologues in Metamonada are predicted to function in the cytosol, with the only exception of the ACS2 isoform from *S. salmonicida* that functions in the MRO [5,56]. Phylogenetic analysis of Preaxostyla ACSs (Fig E in S4 File) shows four unrelated clades, none in close relationship to the *S. salmonicida* MRO homolog, hinting at a cytosolic localization of these enzymes in Preaxostyla.

## Amino acid metabolism is richer in *Trimastix* and *Paratrimastix* than in oxymonads

We noticed clear differences between oxymonads on one side and MRO-containing species *T. marina* and *P. pyriformis* on the other with respect to their amino acid metabolism.

Hypothetical amino acid metabolism was reconstructed based on the metabolic maps in the KEGG database and catalytic activities of enzymes reported from other metamonads (S1–S3 Figs; [57,58]). The capacity to *de novo* synthesize protein-building amino acids seems larger in *P. pyriformis* (cysteine, serine, glycine, threonine, and selenocysteine) than in oxymonads (cysteine, serine, and selenocysteine only). The inferred capacity of *T. marina* to synthesize only two amino acids (cysteine and selenocysteine) most likely reflects the incompleteness of the sequence data available. The selenocysteine biosynthesis pathway present in *P. pyriformis*, *T. marina*, and *M. exilis* is notable, as the capacity to synthesize this amino acid has been reported only in *S. salmonicida* among other metamonads studied so far [58].

Like many other metamonads [59], *M. exilis* has been suggested to utilize arginine for ATP production via the arginine deiminase pathway consisting of three enzymes: arginine deiminase (ADI), carbamate kinase (CK), and ornithine transcarbamylase (OTC). This important metabolic capability has been probably formed in the common ancestor of Metamonada by the acquisition of genes for ADI and OTC via LGT [59]. Here we show the presence of the complete arginine deiminase pathway also in *T. marina* and *B. nauphoetae*, while *P. pyriformis* and *S. strix* lack ADI and CK, respectively. This suggests that ATP production via arginine catabolism is widespread but not omnipresent in Preaxostyla, and it is present in both free-living and endobiotic representatives. Other amino acids can be used in energy metabolism as well: cysteine, serine, and tryptophan can be converted to pyruvate, while methionine can be converted to α-keto-butyrate. Both products then can be used by PFO and ACS to generate ATP [60].

Of note is the part of amino acid metabolism connected to the folate and methionine cycle. Methionine (in the form of S-adenosylmethionine, SAM), consumed in reactions catalyzed by SAM-dependent methyltransferases, can be regenerated in the methionine cycle, which is present in *P. pyriformis* and *T. marina* but not identified in any of the three oxymonads. The presence of the methionine cycle is probably connected with the presence of MROs containing a complete glycine cleavage system (GCS) and serine hydroxymethyl transferase (SHMT) in *P. pyriformis* [17] and hypothetically also in *T. marina*. The methyl residues liberated from glycine and serine inside MRO enter the folate cycle and can be later utilized for the methylation of homocysteine to produce methionine via MetH [61]. Oxymonads lack the folate and methionine cycles as well as SHMT and GCS, which is in line with the fact that the latter complex is restricted to the mitochondrial compartment in eukaryotes. The transsulfuration pathway, associated with the folate and methionine cycles in mammals [62], was not found in any Preaxostyla species.

Our search for components of the GCS in Preaxostyla led us to identifying a protein that constitutes a potential adaptation to the anaerobic lifestyle more broadly shared by multiple unrelated anaerobic eukaryote lineages. Phylogenetic analysis of homologues of the L protein (GCS-L) divided the sequences from Preaxostyla into two clusters (Fig F in S4 File). The previously characterized MRO-localized protein of *P. pyriformis* considered to function as a bona fide GCS subunit (PAPYR_5544; [17]) branched with sequences of *T. marina*, Parabasalia, and the fornicate *Carpediemonas membranifera*, suggesting that these proteins may be localized in MRO of these species since their common ancestor. Another *P. pyriformis* gene (PAPYR_1328) and its homologues from the oxymonads *M. exilis* and *B. nauphoetae* branched with high statistical support together with sequences from Archamoebae (*Mastigamoeba balamuthi* and *P. schiedti*) and another eukaryotic anaerobe, *Breviata anathema* (Breviatea); we refer to this divergent clade as GCS-L2. Localization of this protein in *P. pyriformis* could not be established as it was measured only in one replicate of the proteomic experiment [17], but given the lack of an N-terminal extension that could potentially serve as a targeting signal into the MRO, we hypothesize it is localized outside this organelle. The well-supported relationship between the GCS-L2 sequences from Preaxostyla, Archamoeabae, and Breviatea may be explained by a eukaryote-to-eukaryote LGT, although the direction of the transfer among the

three eukaryote lineages is unknown. Notably, the *M. balamuthi* GCS-L2 protein was encountered before and suggested to function outside the context of the GCS [63], a notion corroborated here by the identification of its close relatives in oxymonads that lack homologues of the other GCS components. It should be noted that, besides GCS, GCS-L (proper) is known to be part of three other protein complexes, pyruvate dehydrogenase, branched-chain amino acid dehydrogenase, and 2-oxoglutarate dehydrogenase [64], none of them being present in Preaxostyla. Outside of these typical roles or under specific conditions, GCS-L was shown to have a moonlighting proteolytic activity [65] or a diaphorase activity by which it oxidizes NADH using labile ferric iron [66], nitric oxide [67], or ubiquinone [68]. It is, therefore, possible that one of these moonlighting activities may represent the primary role of GCS-L2.

## Oxymonads use almost exclusively 4Fe-4S clusters in their iron-sulfur clusters containing proteins

Fe-S clusters are ancient and versatile inorganic cofactors used by virtually all living organisms and in eukaryotes mostly present in the rhombic 2Fe-2S or cubane 4Fe-4S forms [69]. In a typical eukaryotic cell, Fe-S clusters are synthesized by a combination of the mitochondrial ISC and the cytosolic CIA pathway, but members of Preaxostyla combine the CIA pathway with the SUF pathway acquired by LGT from bacteria to perform the same task [69,70]. The enzymes were shown to localize in the cytosol [17] and the first functional details of the Fe-S cluster biogenesis in this group were described recently [71]. This major pathway shift in the Preaxostyla common ancestor was likely a preadaptation for the loss of the mitochondrion in the lineage leading to *M. exilis* [7].

To assess how much the change in the Fe-S cluster assembly pathway affected the inventory of Fe-S cluster-containing proteins in Preaxostyla, we compared the previously predicted Fe-S proteins of *M. exilis* [8] with a set of Fe-S proteins newly predicted from *in silico* proteomes of *P. pyriformis*, *T. marina*, *B. nauphoetae*, and *S. strix* (S7 File). The numbers of Fe-S cluster-containing proteins identified in individual species varied from 48 in *T. marina* (most likely an underestimate of the real number, owing to the incompleteness of the data) to 93 in *P. pyriformis* and are thus not decreased in comparison to other heterotrophic eukaryotes [72]. The predicted Fe-S proteins fell into 164 distinct orthologous groups. The numbers of these OGs shared by different combinations of Preaxostyla species are shown in S4 Fig. The most widespread types of Fe-S cluster-containing proteins in Preaxostyla fall into expected functional groups acting in processes of extended glycolysis and electron transfer (PFO, HydA, ferredoxins, and flavodoxin-ferredoxin domains containing proteins), DNA replication and repair, transcription and translation (e.g., DNA and RNA polymerases, Rli1p), Fe-S cluster assembly itself (SufDSU, SufB, Nbp35, and NAR-1), and nucleotide (xanthine dehydrogenase, XDH) and amino acid metabolism (L-serine dehydratase).

Most of the predicted Fe-S proteins in oxymonads contain 4Fe-4S clusters, with the single exception of XDH, which contains 2Fe-2S clusters. The free-living *P. pyriformis* and *T. marina* furthermore contain other proteins with 2Fe-2S clusters, such as [FeFe] hydrogenases and 2Fe-2S ferredoxin. As the 2Fe-2S clusters occur more frequently in mitochondrial proteins, the higher number of 2Fe-2S proteins in *P. pyriformis* compared to the oxymonads may reflect the presence of the MRO in this species and its absence in oxymonads.

## Peroxisomes and pathways for the synthesis of heme are absent in all Preaxostyla

Besides modifications or a complete loss of mitochondria, living in low-oxygen environments and/or living inside hosts typically also impacts peroxisomes. While in most anaerobic/

microaerophilic species, including all previously investigated Metamonada, the peroxisomes are absent, both free-living and parasitic Archamoebae recently turned out to possess modified versions of this organelle [73–75], as does *Proteromonas lacertae*, a stramenopile that possesses the most reduced but seemingly functional peroxisomal organelle yet reported [76]. Specific genetic markers for the presence of peroxisomes are peroxins, a set of proteins involved in peroxisome biogenesis [77]. Peroxins are absent in the genome of the oxymonad *M. exilis* with a single exception of a divergent homologue of Pex19 [8]. This protein acts as a soluble cytoplasmic receptor for peroxisomal tail-anchored (TA) membrane proteins, which are inserted into the peroxisomal membrane with the help of the Pex3 anchor. The absence of a discernible Pex3 homologue in *M. exilis* indicates that the Pex19 homologue is involved in another cellular process. To map the situation in other Preaxostyla, we searched all datasets for homologues of human peroxins (S8 File). Besides putative homologues of Pex19, which were recovered in all species besides *S. strix*, and a Pex3 homologue recovered in *P. pyriformis*, no other peroxins were detected. The Pex3/Pex19 pair in *P. pyriformis* was already reported by Zítek et al. [17], who provided proteomic evidence for Pex3 being localized in the MRO and hypothesized that the Pex3/Pex19 system is involved in the targeting of TA proteins into the outer MRO membrane. Altogether, the results of our searches do not indicate the presence of peroxisomes in Preaxostyla lineage. Thus, the apparent absence of Pex3 in oxymonads seems to be another trait linked to the loss of the mitochondrion in these organisms.

The modification of mitochondria for life in low-oxygen environments typically leads to the loss of the capacity for the synthesis of heme [78], a cofactor of many proteins involved in redox reactions, namely the complexes of the respiratory chain. Preaxostyla are no exception in this respect as no homologues of these enzymes were identified in their datasets. Still, cytochrome b5 domain-containing proteins were recovered in all Preaxostyla (S6 File) indicating that they use hemoproteins and likely sequester heme from food.

## Membrane transporter complement may reflect adaptation to an endobiotic lifestyle in oxymonads

The collected -omic data set from Preaxostyla allows us to examine two other evolutionary changes besides the loss of the mitochondrion, the transition from a free-living to an endobiotic lifestyle and the loss of stacked Golgi. Although all three events occurred on the same branch of the phylogenetic tree, leading to the common ancestor of the three analyzed oxymonads, we have no clear evidence that they are directly related. However, an indirect link between them is possible. For example, the transition to an endobiotic mode of life could cause the loss of the methionine cycle in oxymonads. This change might subsequently allow for the loss of mitochondria, whose only essential role in free-living Preaxostyla species is to provide methyl groups for this cytosolic cycle [17].

To assess genetic trace of the transition to endobiotic lifestyle, we examined proteins responsible for the transport of metabolites and other chemicals across the plasma membrane and other cell membranes. We searched for a broad spectrum of transmembrane transporters (except for transporters of MROs) using homology-based methods in order to compare the repertoire of functional types as well as the number of paralogues between the five species of Preaxostyla (S9 File).

The most gene-rich group of membrane transporters identified in Preaxostyla is the ATP-binding cassette (ABC) superfamily represented by MRP and pATPase families, just like in *T. vaginalis* [57]. Altogether, representatives of 19 transporter families have been identified in Preaxostyla. All of them are present in both free-living species *P. pyriformis* and *T. marina*, while four families (PotE, SPNS, RFC, and TauE) are missing in all three endobiotic

oxymonads investigated. On the other hand, transporters of nucleosides, sugars, amino acids, choline, and phospholipids have consistently higher numbers of paralogues in the genomes of the oxymonads *M. exilis* and *B. nauphoetae* than in *P. pyriformis* and their phylogenetic trees are consistent with gene family expansion in oxymonads (Figs G–K in S4 File). These two observations parallel the findings of functional domain loss and expansion by gene duplication of transporter families in Microsporidia [79], possibly hinting at a broader evolutionary pattern at the transition to the endobiotic lifestyle. Indeed, gene duplication in specific types of transmembrane transporters may be a common adaptation at the transition to a new environment in eukaryotes; for example, expansion in gene families encoding ion transporters has been observed in halophilic organisms [80–82] when compared to their mesophilic relatives.

### Evidence from the Golgi complex-associated protein complement in Preaxostyla is consistent with the cisternal adhesion model of Golgi stacking

Another evolutionary transition addressable by our results is the presence of a morphologically identifiable Golgi body in *P. pyriformis* and *T. marina* versus the lack in oxymonads [14,21,83]. This absence of a morphologically identifiable Golgi-homologue and yet the detection of a substantial complement of proteins associated with Golgi membrane-trafficking and transport was used in the initial Karnkowska et al. 2016 paper to provide evidence for a Golgi body, in the oxymonad *M. exilis* [7,8], serving as a positive control for the informatic ability to detect genomic signals of cryptic organelles. Golgi bodies with non-canonical morphologies are relatively rare in eukaryotes, as compared with the hallmark stacked cisternal form [84]. However, such organelles are well-documented [84]. Surprisingly, the molecular basis for the canonical eukaryotic stacked morphology is unclear. Many proteins have been implicated in Golgi stacking, though none yet found to be universal. At the same time the cisternal adhesion model [85], an emerging but still relatively controversial theory, proposes that a sufficient amount of any one of several putative stacking factors could provide the adhesive property to keep the stacks together. A 2018 study by Barlow et. al. [86], looked for all proposed Golgi-stacking factors at the time by sampling genomic data from across eukaryotes possessing stacked and unstacked Golgi morphologies. No protein was found in a pattern that supported it as a necessary and sufficient for Golgi stacking, although the complement of Golgins inferred to be present in LECA did support the functional distinction of cis vs trans-Golgi. This finding was most consistent with the cisternal adhesion model. Nonetheless, the organisms sampled with divergent Golgi morphologies were still quite distantly related and thus the evidence supporting the model was still relatively weak.

By contrast, our dataset represents one of the tightest samplings to date where genomic data is available for the closest known species on both sides of the divide between stacked and unstacked Golgi morphology. Consequently, we searched in our datasets to assess the Golgi complement in the additional Preaxostyla representatives, particularly to see whether the complement was more extensive in the organisms possessing stacked Golgi bodies.

The scope of our examination included proteins involved in vesicle formation, vesicle fusion, and the golgin proteins implicated in Golgi structure [87–92]. We observed near complete complements of the COPI, AP1, AP3, AP4, and Retromer vesicle coats, as well as the GARP complex, Trs120, and syntaxins 5 and 16 (Fig 4 and S6 File). We also noted at least one golgin protein in each of the organisms. Indeed, we observed additional paralogues of the vesicle trafficking machinery (e.g., AP1, Retromer, GARP/EARP) in oxymonads compared to *P. pyriformis* and *T. marina*, (Fig 4 and S6 File). These data, together with previously published observations [93], are indicative of Golgi bodies with multiple anterograde and retrograde pathways entering and exiting the organelle present in all Preaxostyla species.

We did observe two clear differences in the sets of Golgi-associated proteins between the stacked and unstacked Golgi-possessing organisms. Firstly, *P. pyriformis* encodes seven of the eight Conserved Oligomeric Golgi (COG) complex proteins, while only a sparse representation of the COG complex was seen in the oxymonads. Secondly, the golgin CASP was found in both *P. pyriformis* and *T. marina*, but in none of the oxymonad genomes. The same was true for Golgin 84. This marks the first report of CASP or Golgin 84 from a metamonad [86] suggesting independent losses of these proteins in the Oxymonadida, Parabasalia, and Fornicata lineages. While caution is warranted when reporting the absence of any given single protein from any given genome, our observations do show a greater number of encoded Golgi-structure implicated proteins in the stacked-Golgi possessing lineages than in the oxymonads (4, 7 vs 1, 4, 3 respectively; Fig 4). Though expression levels would need to be taken into account, this observation is nonetheless consistent with the cisternal adhesion model [85], i.e., that it is the amount of adhesive golgin-type proteins that regulate stacking rather than the identity of any given Golgi-stacking protein [86].

## Conclusions

In this manuscript we report on the results of thorough searches for mitochondrion-associated proteins, which failed to uncover any convincing candidates in the three investigated oxymonads but corroborate the presence of a unique MRO type in *T. marina* and *P. pyriformis*. This shows that the absence of mitochondria is not a unique feature of a single species, *M. exilis*, but applies to a wider range of oxymonads, putatively the whole group. This fact moves this unique loss back to at least 100 MYA, when oxymonads had been already diversified [94]. It also implies that a eukaryotic lineage without mitochondria can thrive for a long time and undergo pronounced morphological evolution, as is apparent from the range of shapes and specialized cellular structures exhibited by extant oxymonads [10]. We also noticed that the loss of mitochondrion has not greatly affected the counts of Fe-S cluster-containing proteins but led to a decrease in the usage of 2Fe-2S cluster types. On the other hand, it might relate to the loss of large hydrogenases in oxymonads and the simplification of their amino acid metabolism, which lacks the glycine cleavage system, a connection to one carbon pool. The latter is considered the only essential function of MRO in *P. pyriformis* [17] and its removal was probably the last step towards amitochondriality.

We also identified differences in inventories of membrane transporters between free-living *P. pyriformis* and *T. marina* and oxymonads which we ascribe to the transition from the free-living to the endobiotic lifestyle at the origin of oxymonads. Finally, proteins involved in the formation and regulation of the Golgi structure have a patchy distribution and show a trend towards loss in oxymonads, which is consistent with the lack of ultrastructural evidence for the presence of a stacked Golgi in oxymonads, but also with an emerging cell biological model for how Golgi maintain their hallmark morphology of stacked cisternae [85].

## Materials and methods

### Cell culture, DNA and RNA isolation

Monoeukaryotic, xenic culture of *P. pyriformis* (strain RCP-MX, ATCC 50935) was maintained in the Sonneborn's Paramecium medium ATCC 802 at room temperature by serial transfer of one ml of well-grown culture (approximately $5 \times 10^4$ cells/ml) into a 15 ml test tube containing 10 ml of fresh, bacterized medium every week. The medium was bacterized 24 hours before the transfer. *B. nauphoetae* strain NAU3 [21] was cultured in a similar way as described above but using a modified TYSGM media [95].

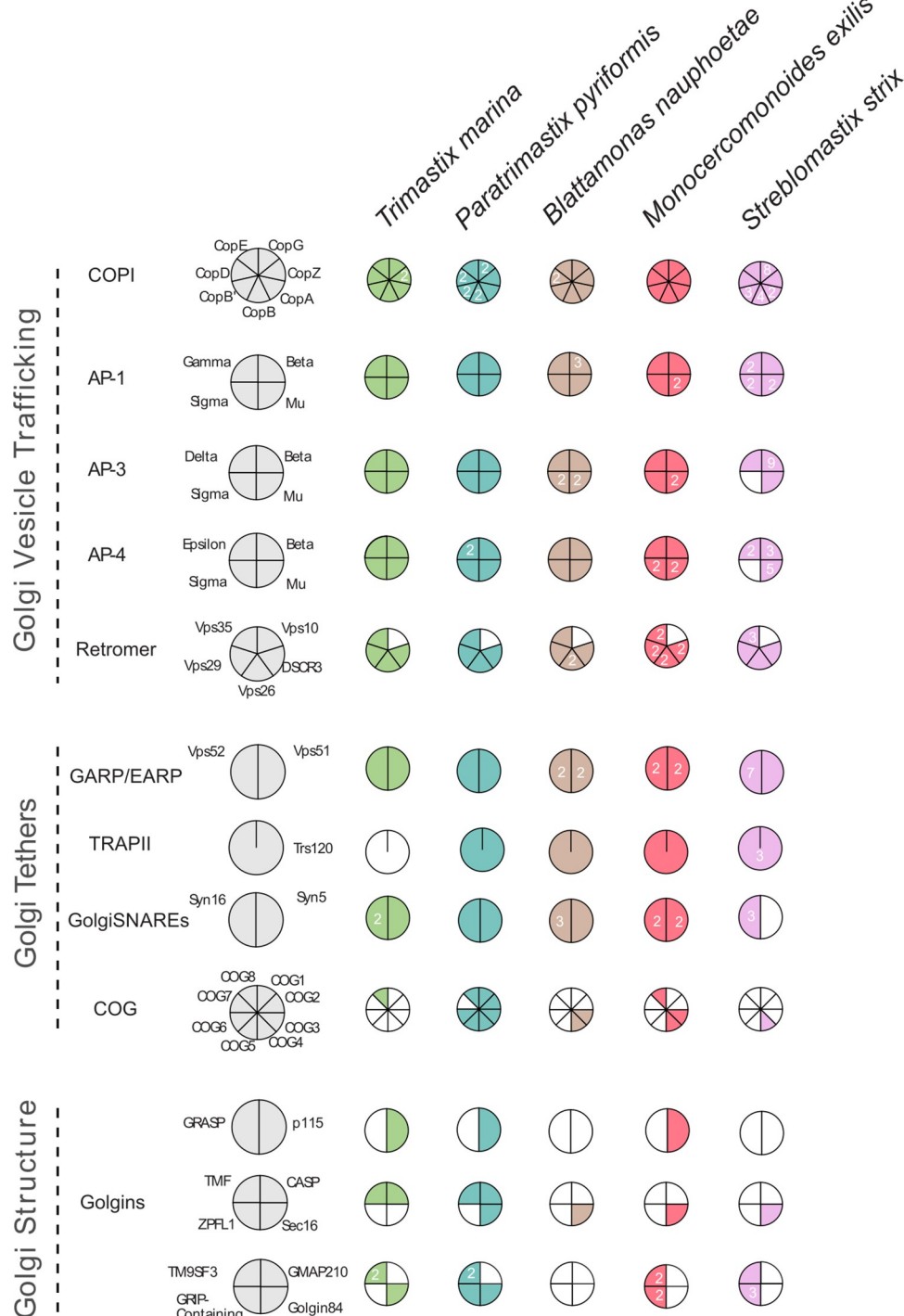

**Fig 4. Complement of Golgi-associated proteins in Preaxostyla.** This Coulson plot shows the set of proteins present in the Preaxostyla predicted proteomes. Empty segments denote failure to identify a candidate orthologue, while filled segments denote success, with paralog numbers inset as relevant. Candidate proteins are identified by homology-searching and verified by phylogenetics as relevant. Details are given in S6 File.

For DNA isolation, 15 liters of *P. pyriformis* and 32 liters of *B. nauphoetae* culture were used. To remove most of the bacterial contamination, the cells were filtered as described previously [7]. After filtration, the cells were collected at 1200×g for 10 min at four ˚C. The DNA was isolated using two different kits. The gDNA samples for PacBio, Illumina HiSeq, and Illumina MiSeq sequencing were isolated using the Qiagen DNeasy Blood & Tissue Kit (Qiagen). The isolated gDNA was further ethanol-precipitated to increase the concentration and remove any contaminants. For Nanopore sequencing, DNA was isolated using Qiagen MagAttract HMW DNA Kit (Qiagen) according to the manufacturer's protocol.

RNA for Illumina transcriptome sequencing of *P. pyriformis* was isolated from 10 liters of a well-grown culture using TRI reagent (Sigma-Aldrich). Eukaryotic mRNA was selected from total RNA using Dynabeads Oligo (dT) beads (Thermo Fisher Scientific). cDNA was synthesized using the SMARTer PCR cDNA Synthesis Kit (Takara Bio Group) and sequenced using the Illumina HiSeq 2000 sequencer at the Beijing Genomics Institute (BGI; Shenzhen, People's Republic of China).

For RNA isolation of *B. nauphoetae*, we used 500 ml of a well-grown culture. Prior to RNA isolation, the culture was filtered as described previously [7], and the cells were harvested by centrifugation at 1200×g for 10 min at four ˚C. The total RNA was isolated using TRI reagent (Sigma-Aldrich) according to the manufacturer's protocol. From the total RNA, mRNA was isolated using two rounds of Dynabeads Oligo (dT) beads (Thermo Fisher Scientific) according to the manufacturer's protocol. Purified mRNA was used for *de novo* whole transcriptome sequencing performed at the Beijing Genomics Institute (BGI; Shenzhen, People's Republic of China).

## Library preparation, Illumina, and Nanopore sequencing

For *P. pyriformis*, PacBio sequencing was performed at SEQme s.r.o. (Dobříš, Czech Republic) using a PacBio RSII sequencer, whereas Illumina sequencing was performed using Illumina HiSeq and MiSeq sequencers at the Institute of Molecular Genetics of the ASCR, v. v. i. (Prague, Czech Republic). For *B. nauphoetae*, one pair-end and two mate-pair libraries were prepared and sequenced on Illumina MIseq PE 2x300bp at Beijing Genomics Institute (BGI; Shenzhen, People's Republic of China).

Libraries for Nanopore sequencing were prepared from four μg of gDNA for each library. DNA was sheared at ~20kbp using Covaris g-TUBES (Covaris Ltd, UK) and the library was prepared using the ligation sequencing kit from Oxford Nanopore Technologies (SQK-LSK108) according to the manufacturer's protocol. The prepared libraries were loaded onto a R9.4 Spot-On Flow cell (FLO-MIN106) and sequencing was performed on a MinION Mk1B machine for 48 hours using MinKNOW 2.0 software with live base calling enabled. In total we prepared four libraries and used four flow-cells for sequencing, two for *P. pyriformis*, and two for *B. nauphoetae*.

## Genome assembly binning and decontamination

The quality of sequencing data was assessed with FastQC (Babraham Bioinformatics, USA). For the Illumina data, adapter and quality trimming were performed using Trimmomatic 0.36 [96], with a quality threshold of 15. For the Nanopore data, trimming and removal of chimeric reads were performed using Porechop v0.2.3 (github.com/rrwick/Porechop).

The initial assembly of the genomes was made only with the Nanopore and PacBio reads using Canu v1.7.1 assembler [97], with corMinCoverage and corOutCoverage set to zero and 100,000 respectively. After assembly, the data was binned using tetraESOM [98]. The resulting eukaryotic bins were also checked using a combination of BLASTn and a scoring strategy based on the identity and coverage of the scaffold as described in Treitli et al. [99]. After

binning, the resulting genomic bins were polished in two phases. In the first phase, the scaffolds were polished with Nanopolish [100] using the raw reads generated by Nanopore. In the second phase, the resulting scaffolds generated by Nanopolish were further corrected using Illumina short reads with Pilon v1.21 [101]. Finally, the genome assembly of *P. pyriformis* was further scaffolded with raw RNA-seq reads using Rascaf [102].

## Repeat masking, gene prediction, and annotation of the genomes

Repetitive elements in the genomic assembly were identified using RepeatModeler v1.0.11 [103], and only repeats that are members of known repeat families were used for masking the genome assemblies prior to gene prediction. For the *P. pyriformis* and *B. nauphoetae*, we used Augustus 3.2.3 for gene prediction [104]. For *de novo* prediction of genes, Augustus was first re-trained using a set of gene models manually curated by considering information from mapped transcriptomic sequences and sequence conservation with homologous protein-coding genes. In the next step, intron hints were generated from the RNAseq data and gene prediction was performed on repeat-masked genomes using Augustus. Next, transcriptome assemblies were mapped to the genomes using PASA [105] and the resulting assembled transcripts were used as evidence for gene model polishing with EVM [106]. The genome completeness for each genome was estimated using BUSCO v3 with the Eukaryota odb9 dataset and the genome completeness was estimated on the sets of EVM-polished protein sequences as the input. Reference-independent genome completeness estimation was performed using Merqury [19] with the Illumina decontaminated reads. To avoid bias in the results, the read decontamination was performed by removing only those reads that mapped to the contaminants identified during genome decontamination.

Automatic functional annotation of the predicted genes was performed using the KEGG Automatic Annotation Server [107], in parallel to similarity searches against NCBI nr protein database using BLASTp. If both gave functional annotation, BLAST was preferred if the e-value was lower than $10^{-30}$. Otherwise KEGG was used. Targeted analyses of genes and gene families of specific interest were performed by manual searches of the predicted proteomes using BLASTp and HMMER [108], and complemented by tBLASTn searches of the genome and transcriptome assemblies to check for the presence of individual genes of interest that were potentially missed in the predicted protein sets (single digits of cases per set). Gene models were manually refined for genes of interest when necessary and possible. The annotated genomes were deposited in NCBI under accession number JAPMOS000000000 for *P. pyriformis* and JARBJD000000000 for *B. nauphoetae*.

## Orthology grouping

We combined the gene inventories of five Preaxostyla species (the two genomes presented here, the *M. exilis* genome GiardiaDB-46_MexilisPA203 in GiardiaDB release 46, the *S. strix* genome under NCBI BioProject PRJNA524138, and the *T. marina* transcriptome-derived predicted proteome in EukProt EP00771, [109]) with 14 other metamonads for which the genome or transcriptome drafts were available when the study was initiated (Sheet A in S3 File). To optimize the inflation parameter value of OrthoMCL, we performed multiple clustering runs with different inflation parameter values in the range 1.01 to 30.0 and calculated the number of OGs containing genes from all Preaxostyla but no other taxa and from all oxymonads but no other taxa. We chose to proceed with the value of inflation parameter = 4, because under this setting the sum of these two numbers was the highest, so the strength of the clustering should be optimal for our purpose. Under this setting, the complete set of proteins from the 19 species (337,300 items) was assigned to 37,923 groups of genes or singletons (OGs) (Sheet B in

S3 File). Proteins belonging to these OGs were automatically annotated using BLASTp searches against the NCBI nr protein database to acquire functional annotation (Sheet C in S3 File). Venn diagrams were generated using InteractiVenn [110].

## Search for mitochondrial proteins

The comprehensive search for putative mitochondrial protein was performed for all five Preaxostyla species, including the previously analyzed *M. exilis* [7] as a control. The general design of the search followed the previously described methodology [7]. Briefly, a custom mitochondrial protein sequence database was compiled using the MitoMiner v4.0 database [22,23] as the core and supplemented with MROs proteins from sixteen different organisms [5,50,51,63,111–116]. Redundant homologues (90% similarity threshold) were removed from the database using CD-HIT. The resulting non-redundant database, herein referred to as MitoMiner, contained 6,979 proteins.

Most searches were divided in three phases. 1) hits generated by HMM searches, predictions of mitochondrial targeting sequence, predictions of transmembrane domains of TA proteins or predictions of mitochondrial β-barrel outer membrane proteins, were in (2) narrowed down to candidates, proteins with hits in the MitoMiner database. 3) All candidates were BLAST-searched against the NCBI nr database and the best hits with the descriptions not including the terms 'low quality protein', 'hypothetical', 'unknown', etc. were kept. Gene Ontology categories were assigned using InterProScan-5.36–75.0. If the annotations received from BLAST or InterProScan corresponded to the originally suggested reliably mitochondrial function, the candidates were considered as mitochondrial proteins.

HMM searches using HMMER 3.1b2 and profile HMMs previously employed by Karnkowska et al. [7] were performed to specifically identify proteins involved in mitochondrial protein import and translocation, as these were shown to be often divergent [51]. Mitochondrial targeting signals were detected using TargetP v1.1 [117] and MitoFates v1.1 [118]. Both programs indicate a probability of mitochondrial localization of the protein. Hits with the probability of mitochondrial localization indicated to be >0.5 by both programs were considered for manual verification. To find tail-anchored proteins, transmembrane domains (TMDs) were predicted using TMHMM2.0 [119] for all analyzed proteins. Hits with a TMD within 32 amino acid residues from the C-terminus were kept for verification. The mitochondrial β-barrel outer membrane proteins (MBOMPs) search has been conducted using the pipeline described by Imai et al. [120]. The pipeline firstly identifies a β-signal ($P_oxGh_yxH_yxH_y$ motif), required for the insertion into the membrane, in the C-terminus of the query protein. Subsequently, the secondary structure of the stretch of 300 amino acid residues preceding the β-signal is analyzed using PSIPRED [121] to check for a typical β-structure. Significant hits, with at least 25% of the sequence predicted to form β-strands, no more than 10% assumed by an α-helical structure, and no more than 50% of the eight residues of the β-signal predicted as an α-helical structure, were analyzed further.

On the top of the three-phase searches, we have also performed one extensive but low specificity homology search in which reciprocal BLAST analysis using the MitoMiner database was performed for each predicted proteome of Preaxostyla species with an e-value threshold of 0.001. Functional annotation of the reciprocal hits was assessed by NCBI BLAST and items were inspected for cases of reliable mitochondrial proteins.

## *Trypanosoma brucei* mitoproteome-guided comparative analyses

Predicted proteomes of *T. marina*, *P. pyriformis*, *S. strix*, *B. nauphoetae*, and *M. exilis* were individually reverse BLAST-searched against the proteome of *T. brucei* (downloaded from

Tritrypdb.org; November 2019; [122]). Only reciprocal best BLAST hits that were identified in any of the previously published mitochondrial proteomes of *T. brucei* [37,39] were subjected to further phylogenetic analysis. For each such protein, the data set for the tree construction was composed of hits from a custom BLAST database of selected protist proteomes (downloaded from UniProt, November 2019; [123]). Protein sequence sets were automatically aligned with MAFFT v7.453 using the L-INS-i refinement and a maximum of 1,000 iterations, followed by trimming of sites with >70% gaps. ML trees (S5 File) were inferred by IQ-TREE v 1.6.12 using the Posterior Mean Site Frequency (PMSF) empirical model with the ultrafast bootstrapping strategy (1,000 replicates) and a LG4X guide tree [124]. Subcellular targeting of all proteins in the tree was predicted by using TargetP-1.1; the presence of a signal peptide, a chloroplast targeting peptide or a mitochondrial targeting peptide in the respective proteins is marked by the letters S, C, or M, respectively, at the very beginning of the sequence names.

### Prediction of proteins containing Fe-S clusters

Fe-S cluster containing proteins were predicted with MetalPredator webserver [125]. Predicted proteins were functionally annotated by BLAST searches against the NCBI nr database and by InterProScan against the InterPro database [126]. KEGG categories were assigned by Ghost-KOALA searches against the KEGG database [127]. Orthologous groups were created with the OrthoMCL software [128] and the Venn diagram of OG sharing among Preaxostyla was visualized using InteractiVenn.

### Search for Peroxins and enzymes for the heme synthesis

The peroxins and the proteins involved in heme biosynthesis were identified by BLASTp searches. All BLASTp hits with an e-value of less than 0.1 were further analyzed by reciprocal BLASTp and HHpred [129].

### Single gene phylogenies

The phylogenies of the genes of interest were analyzed individually using the methodology described below. Eukaryotic homologues of the Preaxostyla genes were gathered by taxonomically constrained iterative BLAST search against publicly available sequence databases in order to sample as broad eukaryotic diversity as possible. In the cases of overrepresented taxa of low interest (e.g. Metazoa, land plants), only a small number of representative sequences were selected arbitrarily. In order to detect potential LGT from prokaryotes while keeping the number of included sequences manageable, prokaryotic homologues were gathered by a BLASTp search with each eukaryotic sequence against the NCBI nr database with an e-value cutoff of $10^{-10}$ and max. 10 target sequences. The sequences were aligned using MAFFT and automatically trimmed using trimAl v1.2 [130]. Phylogenetic analyses were performed simultaneously using IQ-TREE v2.0.5 [124] and RAxML-HPC2 on v8.2.12 [131] with the LG4X model on the Cyberinfrastructure for the Phylogenetic Research (CIPRES) Science Gateway [132]. Substitution models were inferred using IQ-TREE TESTNEW function for IQ-TREE.

### Phylogenetic analysis of Golgi-related proteins

Comparative genomics was carried out using HMM searches. Pan-eukaryotic query sequences analyzed and curated for previous pan-eukaryotic vesicle coat, multisubunit tethering complexes, and golgins were used to build profile HMMs. Query sequences were obtained from the supplementary material of previous papers dealing specifically with Adaptins and COPI [133], multisubunit tethering complexes [134], and golgins [86]. Individual components and

proteins from each sub-complex were aligned using MUSCLE v.3.8.31 and the resulting alignment files were used to generate HMM matrices using the HMMBUILD option available through the HMMER package. HMMER searches were carried out in the predicted proteomes of *P. pyriformis*, *B. nauphoetae*, and *S. strix*, whereas for *T. marina* the transcriptome assembly was first translated in all six open reading frames using the *ab initio* gene prediction program GeneMarkS-T using the default parameters [135] and the longest resulting predicted protein sequences were used for the searches. Forward hits meeting an e-value cutoff of 0.05 were subject to reciprocal BLASTp analyses against the *Homo sapiens* and *M. exilis* predicted proteomes as well as the NCBI nr database. Any absent components were also subject to additional tBLASTn searches in the nucleotide scaffolds. Hits were deemed positive if both forward hits and at least two of three reciprocal BLAST results retrieved the correct orthologue with an e-value cutoff of 0.05. Paralogous gene families were subject to further phylogenetic analyses.

Phylogenetic analyses were undertaken for the large, medium, and small subunits from identified AP1-4 and CopI families, as well as Syntaxin16 and Syntaxin5. Resolved backbone alignments from the dataset curated for Karnkowska et al. [8] were used and sequences from *T. marina*, *P. pyriformis*, *B. nauphoetae*, and *S. strix* were iteratively aligned with the backbone alignment using the profile option available through MUSCLE v3.8.31 [136]. All alignments were subsequently inspected and manually trimmed using Mesquite v. 3.5 (mesquiteproject.org) to remove heterogeneous regions lacking obvious homology and partial or misaligned sequences.

Maximum likelihood analyses were carried out using RAxML-HPC2 on XSEDE v.8.2.10 for non-parametric bootstrap replicates [137]. The best protein matrix model for RAxML was tested using ProtTest v.3.4.2 [138], set to account for Gamma rate variation (+G), invariant sites (+I), and observed frequency of amino acids (+F) with default tree and a consensus tree was obtained using the Consense package, available through the Phylip v.3.66 [139].

Bayesian inference was carried out using MRBAYES on XSEDE v3.2.6 [140]. Parameters specified included 10 million Markov Chain Monte Carlo generations under a mixed amino acid model with the number of gamma rate categories set a 4. The sampling frequency was set to occur every 1000 generations with a burn-in of 0.25. Tree convergence was ensured with the average standard deviation of split frequency values below 0.01. Both RAxML and MRBAYES analyses were performed on the CIPRES webserver. RAxML bootstrap values as well as Bayesian posterior probabilities were overlaid on the best Bayes topology with combined values of >50 and >0.80, respectively, indicating branch support. All alignments are available upon request.

## Supporting information

**S1 Fig. Hypothetical map of the amino acid metabolism in *P. pyriformis*.** Brown color indicates enzymes possibly involved in amino acid biosynthesis pathways. Red color indicates enzymes possibly involved in ATP production. Note that some of the connections between metabolites correspond to the mere transfer of the amino group rather than conversion of the carbon backbone of the molecule (green color). Cyan color is used for remaining enzymes. Abbreviations and Enzyme Commission numbers are given in S6 File.
(TIF)

**S2 Fig. Hypothetical map of the amino acid metabolism in *T. marina*.** Brown color indicates enzymes possibly involved in amino acid biosynthesis pathways. Red color indicates enzymes possibly involved in ATP production. Note that some of the connections between metabolites correspond to the mere transfer of the amino group rather than conversion of the carbon backbone of the molecule (green color). Cyan color is used for remaining enzymes. Abbreviations and Enzyme Commission numbers are given in S6 File.
(TIF)

**S3 Fig. Hypothetical map of the amino acid metabolism in *M. exilis*, *B. nauphoetae*, and *S. strix*.** Brown color indicates enzymes possibly involved in amino acid biosynthesis pathways. Red color indicates enzymes possibly involved in ATP production. Note that some of the connections between metabolites correspond to the mere transfer of the amino group rather than conversion of the carbon backbone of the molecule (green color). Cyan color is used for remaining enzymes. Abbreviations and Enzyme Commission numbers are given in S6 File.
(TIF)

**S4 Fig. Venn diagram showing the distribution of orthologous groups (OGs) of Fe-S cluster-containing proteins among the species of Preaxostyla.** The identity of the OGs and of the component proteins are provided in S7 File.
(TIF)

**S1 File. Genome completeness analyses.**
(XLSX)

**S2 File. Results of searches for mitochondrial candidates in Preaxostyla.**
(XLSX)

**S3 File. Orthologous groups analysis.**
(XLSX)

**S4 File. Phylogenetic trees of selected genes individually discussed in this work.**
(PDF)

**S5 File. Phylogenetic trees of mitochondrial candidates.**
(RAR)

**S6 File. Manual annotation of protein-coding genes in Preaxostyla.**
(XLSX)

**S7 File. Results of searches for FeS-cluster containing proteins in Preaxostyla.**
(XLSX)

**S8 File. Results of searches for peroxins in Preaxostyla.**
(XLSX)

**S9 File. Results of searches for transmembrane transporters in Preaxostyla.**
(XLSX)

## Acknowledgments

Computational resources were supplied by the project "e Infrastruktura CZ" (e-INFRA LM2018140) provided within the program Projects of Large Research, Development and Innovations Infrastructures.

## Author Contributions

**Conceptualization:** Lukáš V. F. Novák, Laura Eme, Joel B. Dacks, Anna Karnkowska, Marek Eliáš, Vladimír Hampl.

**Data curation:** Lukáš V. F. Novák, Sebastian C. Treitli, Paweł Hałakuc, Shweta V. Pipaliya, Vojtěch Vacek, Ondřej Brzoň, Petr Soukal, Laura Eme, Joel B. Dacks, Anna Karnkowska, Marek Eliáš, Vladimír Hampl.

**Formal analysis:** Lukáš V. F. Novák, Sebastian C. Treitli, Jan Pyrih, Paweł Hałakuc, Shweta V. Pipaliya, Vojtěch Vacek, Ondřej Brzoň, Petr Soukal, Laura Eme, Joel B. Dacks, Anna Karnkowska, Marek Eliáš, Vladimír Hampl.

**Funding acquisition:** Joel B. Dacks, Anna Karnkowska, Marek Eliáš, Vladimír Hampl.

**Investigation:** Lukáš V. F. Novák, Sebastian C. Treitli, Jan Pyrih, Paweł Hałakuc, Shweta V. Pipaliya, Vojtěch Vacek, Ondřej Brzoň, Petr Soukal, Laura Eme, Joel B. Dacks, Anna Karnkowska, Marek Eliáš, Vladimír Hampl.

**Methodology:** Lukáš V. F. Novák, Sebastian C. Treitli, Jan Pyrih, Paweł Hałakuc, Shweta V. Pipaliya, Vojtěch Vacek, Ondřej Brzoň, Petr Soukal, Laura Eme, Joel B. Dacks, Anna Karnkowska, Marek Eliáš, Vladimír Hampl.

**Project administration:** Lukáš V. F. Novák, Vladimír Hampl.

**Resources:** Vladimír Hampl.

**Supervision:** Laura Eme, Vladimír Hampl.

**Validation:** Sebastian C. Treitli, Laura Eme, Joel B. Dacks, Anna Karnkowska, Marek Eliáš, Vladimír Hampl.

**Visualization:** Lukáš V. F. Novák, Shweta V. Pipaliya, Vojtěch Vacek, Anna Karnkowska.

**Writing – original draft:** Lukáš V. F. Novák, Laura Eme, Joel B. Dacks, Anna Karnkowska, Marek Eliáš, Vladimír Hampl.

**Writing – review & editing:** Lukáš V. F. Novák, Sebastian C. Treitli, Jan Pyrih, Paweł Hałakuc, Shweta V. Pipaliya, Vojtěch Vacek, Ondřej Brzoň, Petr Soukal, Laura Eme, Joel B. Dacks, Anna Karnkowska, Marek Eliáš, Vladimír Hampl.

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
