## [Decision Letter · Decision Letter 0]

24 Aug 2023

Dear Dr Novák,

Thank you very much for submitting your Research Article entitled 'Genomics of Preaxostyla Flagellates Illuminates Evolutionary Transitions and the Path Towards Mitochondrial Loss' to PLOS Genetics.

The manuscript was fully evaluated at the editorial level and by independent peer reviewers. The reviewers appreciated the attention to an important problem, but raised some substantial concerns about the current manuscript. Based on the reviews, we will not be able to accept this version of the manuscript, but we would be willing to review a much-revised version. We cannot, of course, promise publication at that time.

There are several concerns.  The first is the writing style.  It is very difficult to read and the exciting point of the loss of the mitochondria in two more members of the oxymonadida is obscured.  The reviewers did not have many problems with the science, but with the presentation.  The authors should present the data that support the main pint of the paper and put much of the other information in another manuscript. One possible route would be those suggested by Reviewer 3. As written, the manuscript is not acceptable for the general audience of PLoS Genetics.  There some suggestions about rewriting, but the authors will need to dramatically shorten it.

Second, two more oxymonads do not allow the authors to state that all of the oxymonadia have lost mitochondria.  They need to be more restrained with this conclusion.  

If you decide to revise the manuscript for further consideration at PLOS Genetics, please aim to resubmit within the next 60 days, unless it will take extra time to address the concerns of the reviewers, in which case we would appreciate an expected resubmission date by email to plosgenetics@plos.org.

We are sorry that we cannot be more positive about your manuscript at this stage. Please do not hesitate to contact us if you have any concerns or questions.

Yours sincerely,

Susan K. Dutcher

Academic Editor

PLOS Genetics

Eva Stukenbrock

Section Editor

PLOS Genetics

Reviewer's Responses to Questions

**Comments to the Authors:**

Reviewer #1: Novák and colleagues present and excellent study: rich in detail and meticulously carried out.

I agree with the previous reviewers that there is a lot in this paper -- but this is a clear strength (except for that it took me quite some time to digest it all)!

Since this is already a revision, I will not add additional minor request here and there, but rather get to the point: the authors have done a great job of putting together a detailed comparative genomic investigation that not only counts some LGTs, but provides rich insights into exciting biochemistry of these fascinating organisms.

Therefore I will refrain from making any other suggestions -- but one: please add a cladogram to Figure 1 to highlight for the general reader exactly *where* on the tree of eukaryotes we are (most readers of PLOS Genetics will not know what metamonads are)

Reviewer #2: see attached pdf

Reviewer #3: Novák and colleagues present here a very interesting and important manuscript that describes the genomes of two Preaxostyla flagelleates, Paratrimastix pyriformis and Blattamonas nauphoetae. The 2016 manuscript from this group describing the complete loss of a mitochondrial organelle from Monocercomonoides was monumental, and this manuscript goes on to show that at least one more oxymonad has apparently lost MROs, opening the possibility that all oxymonads are amitochondriate.

Overall, I congratulate the authors on the overall quality of the work presented in this manuscript. I do not have major concerns about the methods employed in making the most important conclusions of the manuscript. The completeness of the genomes (central to the arguments about mitochondria) is reasonably good (though minor comments later on), and the bioinformatics searches for LGTs, and Golgi/MRO proteins were rigorous and well thought out. It will be very interesting to learn, once additional species are investigated, whether all oxymonads are amitochondriate, or if Monocercomonoides and Blattamonas are just two instances of mitochondrial loss!

In contrast to my positive feelings on the quality of the work done here, I feel that too many - and somewhat disconnected - stories are being told. The net effect of this is that the impact of your biggest point (about MRO loss) is being blunted. I understand that it's hard to leave out interesting details of your analyses, but the manuscript is very long, and almost feels like it could be two separate stories (core proteomes/LGT vs MRO). I bring this up mainly because I feel that the main conclusions presented here really are important.

Some very minor points:

Line 83 - while I'm enthusiastic about the results, I'm not sure that such a limited number of genomes can yet answer if oxymonads are amitochondriate 'in general'

Line ~100 (genome completeness analysis). I understand that protist genomes often do not give a near 100% score in BUSCO/CEGMA because many universal proteins aren't universal. One additional strategy could be looking for all cytosolic ribosomal proteins; these are highly conserved, typically not lost, and would be readily identifiable from transcriptome data if they aren't present in the genome, as they are highly expressed. Another thought might be looking to see what proportion of transcripts aren't represented in the genome assembly.

Line ~146 - not a critique but a question - are there any consistent signals amongst these LGTs that would give hints about previous symbiotic partners?

Altogether, this manuscript provides a lot of interesting new data, and convincingly argues for the lack of mitochondria in all oxymonads with genomes sequenced to this point.

**Have all data underlying the figures and results presented in the manuscript been provided?**

Reviewer #1: Yes

Reviewer #2: Yes

Reviewer #3: Yes

PLOS authors have the option to publish the peer review history of their article (what does this mean?). If published, this will include your full peer review and any attached files.

Reviewer #1: No

Reviewer #2: No

Reviewer #3: No

---

## [Editor Report · Decision Letter 1]

3 Nov 2023

Dear Dr Novák,

We are pleased to inform you that your manuscript entitled "Genomics of Preaxostyla Flagellates Illuminates the Path Towards the Loss of Mitochondria" has been editorially accepted for publication in PLOS Genetics. Congratulations!  You have really changed the manuscript and made it a more compelling article.

Yours sincerely,

Susan K. Dutcher

Academic Editor

PLOS Genetics

Eva Stukenbrock

Section Editor

PLOS Genetics

Comments from the reviewers (if applicable):

**Data Deposition**

http://datadryad.org/submit?journalID=pgenetics&manu=PGENETICS-D-23-00743R1

**Press Queries**

---

## [Editor Report · Acceptance letter]

14 Nov 2023

PGENETICS-D-23-00743R1 

Genomics of Preaxostyla Flagellates Illuminates the Path Towards the Loss of Mitochondria 

Dear Dr Novák, 

We are pleased to inform you that your manuscript entitled "Genomics of Preaxostyla Flagellates Illuminates the Path Towards the Loss of Mitochondria" has been formally accepted for publication in PLOS Genetics! Your manuscript is now with our production department and you will be notified of the publication date in due course.

With kind regards,

Anita Estes

PLOS Genetics

On behalf of:
